# Neurovascular Coupling Impairment in Heart Failure with Reduction Ejection Fraction

**DOI:** 10.3390/brainsci10100714

**Published:** 2020-10-07

**Authors:** Ana Aires, António Andrade, Elsa Azevedo, Filipa Gomes, José Paulo Araújo, Pedro Castro

**Affiliations:** 1Department of Neurology, Centro Hospitalar Universitário São João, Alameda Prof. Hernâni Monteiro, 4200-319 Porto, Portugal; ana.aires@chsj.min-saude.pt; 2Department of Clinical Neurosciences and Mental Health, Faculty of Medicine of University of Porto, Alameda Prof. Hernâni Monteiro, 4200-319 Porto, Portugal; antoniolfandrade@gmail.com; 3Cardiovascular Research and Development Unit, Alameda Prof. Hernâni Monteiro, 4200-319 Porto, Portugal; eazevedo@med.up.pt; 4Department of Internal Medicine, Centro Hospitalar Universitário São João, Alameda Prof. Hernâni Monteiro, 4200-319 Porto, Portugal; filipa.gomes@chsj.min-saude.pt; 5Department of Medicine, Faculty of Medicine of University of Porto, Alameda Prof. Hernâni Monteiro, 4200-319 Porto, Portugal; jparaujo@med.up.pt

**Keywords:** heart failure, cerebral autoregulation, neurovascular coupling, transcranial Doppler

## Abstract

The hemodynamic consequences of a persistent reduced ejection fraction and unknown cardiac output on the brain have not been thoroughly studied. We sought to explore the status of the mechanisms of cerebrovascular regulation in patients with heart failure with reduced (HFrEF) and recovered (HFrecEF) ejection fraction. We monitored cerebral blood flow velocity (CBFV) with transcranial Doppler and blood pressure. Cerebral autoregulation, assessed by transfer function from the spontaneous oscillations of blood pressure to CBFV and neurovascular coupling (NVC) with visual stimulation were compared between groups of HFrEF, HFrecEF and healthy controls. NVC was significantly impaired in HFrEF patients with reduced augmentation of CBFV during stimulation (overshoot systolic CBFV 19.11 ± 6.92 vs. 22.61 ± 7.78 vs. 27.92 ± 6.84, *p* = 0.04), slower upright of CBFV (rate time to overshoot: 1.19 ± 3.0 vs. 3.06 (4.30) vs. 2.90 ± 3.84, *p* = 0.02); *p* = 0.023) and reduced arterial oscillatory properties (natural frequency 0.17 ± 0.06 vs. 0.20 ± 0.09 vs. 0.24 ± 0.07, *p* = 0.03; attenuation 0.34 ± 0.24 vs. 0.48 ± 0.35 vs. 0.50 ± 0.23, *p* = 0.05). Cerebral autoregulation was preserved. The neurovascular unit of subjects with chronically reduced heart pumping capability is severely dysfunctional. Dynamic testing with transcranial Doppler could be useful in these patients, but whether it helps in predicting cognitive impairment must be addressed in future prospective studies.

## 1. Introduction

Increased rates of cognitive decline and dementia have been documented in patients with heart failure (HF) [1]. From a classical perspective, this could be explained by the fact that HF is linked to increased prevalence of atrial fibrillation and vascular risk factors and, therefore, to a higher incidence of stroke. However, HF also reduces global blood flow, decreasing the oxygen transport to organs, including brain [1], contributing to cerebrovascular injury [2] and autonomic and neuropsychological disturbances [3]. HF repercussion can be assessed by measuring variables that reflect pump effectiveness, such as ejection fraction (EF) and cardiac output (CO). Some previous studies adopted EF, the percentage of blood volume ejected in one cardiac cycle, to infer the severity of the disease while others focused on CO, the volume of blood pumped by the heart in every minute, since it considers both systolic and diastolic functions. These parameters are commonly related but not always in accordance when cardiac function is studied. Independently of how heart performance was estimated, while stroke and cerebrovascular disease has been investigated, the hemodynamic consequences of a persistent reduced EF with unknown CO on the brain have not been thoroughly studied.

A major regulatory mechanism of cerebral blood flow is cerebral autoregulation (CA), the ability to maintain a stable cerebral blood flow despite changes in systemic blood pressure [4]. Impairment of cerebral regulation and cerebral hypoperfusion has been implicated either in relevant cognitive deficits [3,5] and brain structural abnormalities in HF patients [6]. Another crucial aspect of cerebral blood flow control is neurovascular coupling (NVC). This refers to the cerebral blood flow augmentation induced by neuronal activity. The latter is of particular interest because it involves the neurovascular unit (the connection of astrocytes-neurons-capillaries), which has been recently focused in the investigation of cognitive-related disease processes like hypertension and Alzheimer disease [7]. NVC has not been studied before in HF patients. We can hypothesize that impaired NVC could be one additional factor contributing to cognitive decline in HF individuals.

Fortunately, all cerebral hemodynamic regulation mechanisms, including NVC, can be studied non-invasively with transcranial Doppler [8] and EF can be easily assessed with transthoracic echocardiogram. Therefore, we aimed to study if there is an impairment of cerebrovascular regulatory mechanisms in individuals with HF with reduced ejection fraction (HFrEF) and unknown CO in comparison with patients with HF with recovered ejection fraction (HFrecEF) and with healthy controls.

## 2. Materials and Methods

### 2.1. Study Design

A total of 23 patients from the HF outpatient clinic at our center were invited to participate in the study from October 2017 to March 2020.

Patients were considered appropriate to participate in the study if they fulfilled the inclusion criteria: (1) able to contribute throughout the entire duration of the study, (2) age >18 y, (3) diagnosed with systolic heart failure, (4) followed in expert outpatient clinic.

Exclusion criteria were previous history of stroke, dementia or other central nervous system disease that led to cognitive sequels, inadequate temporal window for transcranial Doppler (TCD), significant stenosis ≥50% in carotid or main cerebral arteries, atrial fibrillation, severe aortic stenosis (mean gradient ≥40 mmHg), mechanic or biologic aortic valvular prothesis and other severe valvular diseases capable of interfering significantly with ejection fraction or associated with high risk of cerebral embolization and an expected survival of less than a year.

We recruited two control groups from the institution with similar age and sex to the HFrEF patients: patients with heart failure with recovered ejection fraction (HFrecEF) and healthy controls without heart or cerebral disease and without vascular risk factors.

The study was approved by the local medical ethics committee and all subjects gave their written informed consent.

### 2.2. Medical Evaluation

Medical history and physical examination were performed in all participants, including measurement of weight and height. Systolic and diastolic blood pressure (BP) was averaged from three measurements in the sitting position with an oscillometric cuff (Omron M6, Japan). Heart rate was determined (HR). Orthostatic hypotension was determined by assessing blood pressure over a 2-min period after standing and was defined as a 20 mm Hg fall in systolic blood pressure or a 10 mm Hg fall in diastolic blood pressure. Participants underwent cervical and transcranial Doppler ultrasound examinations (Affinity 50, Philips Healthcare, Netherlands) to exclude hemodynamically significant cervical or intracranial arterial stenosis.

All heart failure patients underwent a transthoracic echocardiogram within 3 months of cerebral hemodynamic evaluation. Left ventricular ejection fraction was evaluated according to guidelines [9,10], as well as cardiac chambers dimensions, transmitral flow and diastolic dysfunction. Cardiac output was not assessed.

Neuropsychological evaluation of all participants was obtained with the Portuguese version of Montreal Cognitive Assessment (MoCA), performed by certified medical staff. It has been validated in the Portuguese population [11].

### 2.3. Cerebral Hemodynamic Procedure

Evaluations were carried according to the previously described protocol [12]. We used a dim lighted room, with ambient temperature, and subjects were in supine position. They avoided caffeine, alcohol, exercise or vasoactive drugs for at least 12 h before evaluation. CBFV was registered in the M1 segment of the right middle cerebral artery (MCA) and the P2 segment of left posterior cerebral artery (PCA), with 2-MHz TCD probes held with a headframe (Doppler BoxX, DWL, Singen, Germany) [3]. Arterial blood pressure was assessed with Finometer (FMS, Amsterdam, Netherlands). Heart rate was obtained with 3-lead electrocardiogram. Data were synchronized and digitized at 400 Hz with Powerlab (AD Instruments, Oxford, UK) and collected for offline analysis. After resting for 20 min, a 5-min period of resting data was stored for determination of CA indexes. Afterwards, NVC protocol was performed.

#### 2.3.1. Cerebral Autoregulation

To analyze cerebral dynamic autoregulation, a 7-min period of resting data was recorded. Transfer function analysis was used to calculate the gain and phase parameters between the mean BP spectrum and CBFV of MCA and PCA [13].

#### 2.3.2. Neurovascular Coupling

NVC was determined by a visual paradigm that consisted of 10 cycles, each with a resting phase of 20 s (eyes closed) and a stimulating phase with a flickering checkerboard at 10 Hz for 40 s [14]. All cycles were synchronized and averaged. Visual stimulation induces a brisk increment in CBFV which overshoots ~10 s and then promptly maintains a steady-state level [14]. NVC response can be assessed in two ways. At first, we obtained the maximum CBFV variation to determine the overshoot parameter as $\frac{maximum\;CBFV-baseline\;CBFV}{baseline\;CBFV}\times 100\%$; systolic and mean CBFV were used [15]. Afterwards, systolic CBFV curve was displayed to express the dynamics of NVC response in time in respect to the second-order linear equation $G\left(s\right)=\;\frac{K\;\times \left(1+Tvs\right)}{\frac{s2}{\omega 2}+\;2\;\xi *\frac{s}{\omega }+\;1}$ [13]. Gain refers to the relative CBFV difference between baseline/rest stage and steady-state level during visual stimulation. Rate time denotes the initial steepness of the CBFV increase, natural frequency and attenuation express oscillatory features of the system [16].

### 2.4. Statistics

Kolmogorov–Smirnov test was used to assess the normality of the continuous variables. Categorical variables were presented as absolute values and percentages and continuous variables as median (interquartile range) or means (±SD) as indicated. Categorical variables were presented as absolute values and percentages and continuous variables as median (interquartile range) or means (±SD) as indicated. Statistical significance for intergroup differences was assessed by Pearson χ2 test for categorical variables and by Mann–Whitney U test or Student t test as indicated for continuous variables. Repeated-measures ANOVA test was used to compare the cerebral hemodynamic data by group (HFrEF patients vs. HFrecEF patients vs. healthy controls) and arterial territory (PCA vs. MCA). A linear regression analysis was employed to detect an association between NVC parameters and absolute values of MoCA score. Multivariable logistic regression analyses were used to determine which of these parameters could be considered as independent predictors of cognitive decline. Statistical analyses were executed using SPSS Statistics, version 25 (IBM, Armonk, NY, USA). A *p*-value of <0.05 (two-sided) was considered statistically significant.

## 3. Results

We recruited 23 patients with HFrEF, 8 patients with HFrecEF and 13 healthy controls. Baseline characteristics of the study groups are compared in Table 1. Patients and controls had similar systolic and diastolic BP and HR but HFrEF group showed increased BMI (*p* = 0.022). Vascular risk factors were equally distributed in HFrEF patients and HFrecEF controls. Mean MoCA scores were also similar in both groups of HF subjects, although 3 out of 20 patients with HFrEF had a MoCA score compatible with cognitive deficit, with 2 SD below appropriate norms, adjusted to age and education level, but without criteria for dementia [11]. 

Cerebral hemodynamic parameters are compared between HFrEF patients and healthy controls in Table 2. There was no significant difference between the three groups concerning cerebral autoregulation. However, NVC was significantly impaired in HFrEF patients with reduced augmentation of CBFV during stimulation (19.11 (6.92) vs. 22.61 (7.78) vs. 27.92 (6.84), *p* = 0.04), slower upright of CBFV (Rate time to overshoot: 1.19 (3.0) vs. 3.06 (4.30) vs. 2.90 (3.84); *p* = 0.02) and reduced arterial oscillatory properties (natural frequency, 0.17 (0.06) vs. 0.20 (0.09) vs. 0.24 (0.07); *p* = 0.03; attenuation 0.34 ± 0.24 vs. 0.48 ± 0.35 vs. 0.50 ± 0.23, *p* = 0.05). Cerebral autoregulation was preserved. Although not statistically significant, HFrecEF subjects tend to perform worse in NVC task, with reduced overshot, gain and natural frequency comparing to healthy controls.

There was no correlation of HF etiology with the various parameters of NVC (*p* > 0.05) (Appendix A).

## 4. Discussion

Our study examined the status of the cerebral hemodynamics regulatory mechanisms with transcranial Doppler in 23 HFrEF patients, 8 HFrecEF patients and 13 healthy controls and found that patients with HFrEF have significant NVC impairment, while CA seems to remain preserved.

### 4.1. Neurovascular Unit Dysfunction

Brain injury in HF results from the interaction of multiple factors and is not well established. Reduced CBF is one of the main mechanisms, but other aspects contribute, as elevated levels of neurohormones and inflammatory reaction. In previous studies with hypotensive patients, insufficient cerebral perfusion led to NVC impairment [17,18], since hyperemic response mediated by astrocytes diminishes in these conditions [19]. Therefore, in patients with reduced ejection fraction, we may expect that this mechanism also contributes to the damage in NVC in hypoperfused areas. Supporting this hypothesis, it is reported that HF improvement leads to better cognitive function [20,21]. However, white matter lesions and cognitive decline are also present in HFrecEF, highlighting the importance of other aspects like vascular risk factors to cortical damage [2]. Decreased NVC is also present in other cardiovascular diseases associated with HF such as hypertension and atrial fibrillation [22,23]. Furthermore, oxidative stress impairs NVC. As HF is a pro-inflammatory state, it induces further damage in this cerebrovascular hemodynamic mechanism [24,25]. In a recent in vivo study by Adamski, M. G. et al., clinical cognitive impairment was associated with cerebral inflammation in HF in mice [26].

### 4.2. Cerebral Autoregulation

Our HFrEF group showed no differences in CA indexes as compared to healthy controls, which is not in agreement with the study of Caldas et al. that states a reduced dynamic CA in ischemic HF patients with EF ≤45% [27]. However, such contrast can be the result of differences in the protocol. They included HF patients with intracranial and extracranial stenosis that are known to significantly impair CA [28,29,30,31]. In our work, we excluded patients with stenosis to eliminate such contributing factor. Furthermore, the study previously mentioned only included patients with heart failure of ischemic etiology while our work involved HF individuals with essentially alcoholic and idiopathic etiologies. Such difference could have major implications as an underlying ischemic environment is frankly associated with white matter lesions [32] leading to CA impairment [33]. Additionally, Caldas et al. studied patients with more severe HF (classes II and III), while our patients and HFrecEF controls were predominantly classified as class I in NYHA. A previous work from our group showed that HF is associated with a better dynamic CA at ischemic hemisphere within 6 and 24 h after stroke, but this effect does not persist 3 months later [34]. A prospective follow up is needed to determine if HF patients will present deterioration or improvement of CA.

### 4.3. Limitations

This pilot study has some limitations. A noticeable limitation of this study is its small sample size and cross-sectional design. A prospective design could help to depict more accurately the differences between groups, namely in what concerns to CA.

We tried to assess cognitive status using MoCA score, which is validated in Portuguese population [10]. However, as mentioned, we studied a small cohort at one specific point in time, preventing the evaluation of the relation with cerebral hemodynamic parameters, disease duration and cognitive decline. Further follow up of a bigger cohort is needed to understand if HFrEF patients will be prone to cognitive impairment.

Another limitation refers to the absence of assessment of cardiac output (CO), since we only measured ejection fraction with transthoracic echocardiogram. Determining CO and its variations at rest and with exercise would help to clarify the impact of CO on cerebral hemodynamics and needs to be clarified in further studies.

Additionally, we need to consider the influence of drugs on cerebral hemodynamics, but we need a larger population to extract solid conclusions about their effects.

Our work shows that NCV, assessed with transcranial Doppler, may be impaired in patients with HFrEF. However, this technique only evaluates large vessels [35], impeding direct evaluation of cerebral microvasculature that may also impact negatively on NVC. Recent studies enhance the need of innovative tools such as the dynamic vessel analyzer-based approach to study retinal circulation [36,37] and functional near-infrared spectroscopy [38] to be used to determine cerebromicrovascular impairment.

Finally, HF is a multifaceted disease with a large spectrum of severity and comorbidities. Each factor has a specific burden in the cerebrovascular hemodynamics regulation which makes more difficult the analyses of data and the achievement of clear-cut conclusions.

## 5. Conclusions

This study shows that NVC impairment seems to be selectively impaired in HFrEF patients. Other mechanisms of cerebrovascular control are preserved in the first stages of disease, like CA. Future research is warranted to understand how HFrEF development could lead to the dysfunction at the neurovascular unit, which could have an impact on the prevention of cognitive impairment. Moreover, it needs to be clarified if the evolution of HF impairs other cerebrovascular regulatory mechanisms.

## Figures and Tables

**Table 1 brainsci-10-00714-t001:** Comparison of baseline characteristics of heart failure with reduced (HFrEF) patients, recovered (HFrecEF) controls, and healthy controls.

**Baseline Characteristic**		**HFrEF** **(*n* = 23)**	**HFrecEF** **(*n* = 8)**	**Healthy Controls** **(*n* = 13)**	***p* Value *^,^^†^**
Age—years, mean (SD)		60 (8)	58 (15)	60 (20)	0.98
Male—*n* (%)		20 (87)	6 (75)	9 (69)	0.42
BMI—kg/m^2^, mean (SD)		29 (5)	27 (4)	26 (2)	0.02
Systolic BP—mmHg, mean (SD)		125 (23)	133 (19)	122 (13)	0.45
Diastolic BP—mmHg, mean (SD)		71 (15)	74 (6)	70 (9)	0.76
PxxMAP, mean (SD)		63 (184)	16 (22)	8 (4)	0.45
HR—bpm, mean (SD)		64 (12)	64 (8)	71 (11)	0.13
Ejection Fraction—%, mean (SD)		27 (8)	49 (12)	-	0.00
Disease duration—years, mean (SD)		3.70 (3.78)	2.75 (1.75)	-	0.35
Hypertension—*n* (%)		9 (31)	2 (25)	-	0.47
Diabetes Mellitus—*n* (%)		9 (31)	1 (13)	-	0.17
Dyslipidemia—*n* (%)		11 (48)	1 (13)	-	0.08
Chronic Kidney Disease—*n* (%)		5 (22)	1 (13)	-	0.57
Excessive alcohol intake^§^—*n* (%)		5 (22)	3 (38)	-	0.38
Tobacco—*n* (%)		4 (17)	3 (38)	-	0.24
Obstructive sleep apnea—*n* (%)		4 (17)	0	-	0.21
Previous Myocardial Infarction—*n* (%)		5 (22)	0	-	0.15
**Medication—*n* (%)**					***p* Value ^†^**
Beta-blockers	Starting dose	14 (60.9)	4 (50.0)	-	0.86
	Target dose	9 (39.1)	3 (37.5)	-	
ACEI	Starting dose	8 (34.8)	1 (12.5)	-	0.05
	Target dose	4 (17.4)	5 (62.5)	-	
ARB	Starting dose	-	-	-	-
	Target dose	2 (8.7)	-	-	
MRA	Starting dose	7 (30.4)	2 (25)	-	0.96
	Target dose	8 (34.8)	3 (37.5)	-	
Sacubitril/valsartan	Starting dose	3 (13)	1 (12.5)	-	0.44
	Target dose	4 (17.4)	-	-	
Ivabradin	Starting dose	5 (21.7)	1 (12.5)	-	0.55
	Target dose	2 (8.7)	-	-	
**NYHA Class—*n* (%)**					***p* Value ^†^**
I		14 (60.9)	5 (62.5)	-	0.94
II		9 (39.1)	3 (37.5)	-	
**HF Etiology—*n* (%)**					***p* Value ^†^**
Idiopathic DCM		10 (44)	1 (13)	-	0.10
Ischemic		7 (30)	1 (13)	-	
Alcoholic		5 (22)	3 (38)	-	
Tachyarrhythmia		1 (4)	1 (13)	-	
Other		-	2 (25)	-	
		**MoCA Scores—*n* (%)**			***p* Value *^,^^†^**
Mean ± SD		24 ± 4	25 ± 6		0.83
Normal		14 (70)	7 (87.5)	-	0.69
−1SD		1 (5)		-	
−1.5 SD		2 (10)		-	
−2 SD		3 (15)	1 (12.5)	-	

ACEI: angiotensin converting enzyme inhibitors; ARB: angiotensin II receptor blockers; BMI: body mass index; BP: blood pressure; bpm: beats per minute; DCM: dilated cardiomyopathy; HF: heart failure; HR: heart rate; IQR: interquartile range; MoCA: Montreal Cognitive Assessment MRA: mineralocorticoid receptor antagonist; NYHA: New York Heart Association classification; SD: standard deviation; ^§^ excess alcohol intake was considered: for men ≥4 drinks per day, for women ≥3 drinks per day. * *t* test *p* values for comparison of means between groups ^†^ Qui-Square test *p* values comparison of frequencies between groups.

**Table 2 brainsci-10-00714-t002:** Comparison of cerebrovascular regulatory parameters between of HFrEF patients, HFrecEF controls and healthy controls.

		**HFrEF** **(*n* = 23)**	**HFrecEF** **(*n* = 8)**	**Healthy Controls** **(*n* = 13)**	***p* Values ***
Mean FV (cm/s)	MCA	53.64 (14.22)	52.94 (9.11)	44.85 (11.25)	0.35
	PCA	33.19 (8.48)	37.10 (4.79)	26.91 (6.49)	
Mean Pfv (cm/s)	MCA	8.26 (7.86)	5.25 (3.23)	4.40 (3.12)	0.62
	PCA	6.10 (5.28)	4.72 (2.78)	3.07 (2.47)	
		**Cerebral Autoregulation**			
		**HFrEF** **(*n* = 23)**	**HFrecEF** **(*n* = 8)**	**Healthy Controls** **(*n* = 13)**	***p* Values ***
VLF Gain (%/mm Hg)	MCA	1.08 (0.53)	0.93 (0.44)	0.97 (0.37)	0.71
	PCA	1.49 (1.06)	1.19 (0.58)	1.15 (0.51)	
LF Gain (%/mm Hg)	MCA	1.12 (0.64)	0.94 (0.58)	1.48 (0.46)	0.67
	PCA	1.49 (0.83)	1.14 (0.56)	1.76 (0.42)	
VLF Phase (radians)	MCA	0.99 (0.47)	1.04 (0.22)	1.15 (0.50)	0.30
	PCA	1.06 (0.52)	1.32 (0.33)	1.17 (0.38)	
LF Phase (radians)	MCA	0.86 (0.50)	0.86 (0.36)	0.73 (0.26)	0.67
	PCA	0.80 (0.55)	0.97 (0.31)	0.79 (0.31)	
		**Neurovascular Coupling (PCA)**			
		**HFrEF** **(*n* = 23)**	**HFrecEF** **(*n* = 8)**	**Healthy Controls** **(*n* = 13)**	***p* Values ***
Overshoot systolic CBFV (%)		19.11 (6.92)	22.61 (7.78)	27.92 (6.84)	0.04
Gain (%)		13.12 (6.31)	16.70 (8.56)	19.65 (4.73)	0.02
Natural frequency (Hz)		0.17 (0.06)	0.20 (0.09)	0.24 (0.07)	0.03
Attenuation (a.u)		0.34 (0.24)	0.48 (0.35)	0.50 (0.23)	0.05
Rate time (s)		1.19 (3.0)	3.06 (4.30)	2.90 (3.84)	0.02

All values are given in mean ± SD. CBFV: cerebral flow velocity; cm/s: centimeter per second; HF: heart failure; Hz: Hertz; LF: low frequency (0.07–0.2 Hz); MCA: middle cerebral artery; mm Hg: millimeters of mercury; PCA: posterior cerebral artery; VLF: Very-low frequency (0.03–0.07 Hz); * *p* value of repeated-measures ANOVA for the interaction between group variable (HFrEF, HFrecEF and heathy controls) and arterial territory (MCA vs. PCA).

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
