# Peer review of "Neurovascular Coupling Impairment in Heart Failure with Reduction Ejection Fraction"

_brainsci, 2020, doi:10.3390/brainsci10100714_

Round 1

Reviewer 1 Report

Authors aimed to assess reduction in cardiac output on brain outcome, predominantly cognitive function, but also physiologic measures such as autoregulation, neurovascular coupling, and cerebrovascular reactivity.  NVC is particularly of interest as it may be an outcome parameter of low cardiac output and etiologic in cognitive decline.   Subjects were divided into those with reduced EF (27), recovered EF (49) and healthy controls.  Findings: a not statistically significant  trend to abnormal MoCA scores with low EF; NVC was impaired with low EF. but not AR and VR.  No mention as to cardiac output.

Most of the major issues with this study relate to the methods. 

  1. Patients:
    1. Total low EF was 23; recovered EF 8, and HC 13.. Not selected by means of cardiac output as stated in Aims, but by EF.  Reduced EF can, and usually does,  have normal CO at rest.  As they were NYHC class I, they probably had normal CO into at least moderate exertion.  This undermines the stated thesis.
    2. Duration of condition is an important variable, particularly for conditions that take time to develop such as cognitive decline.
  2. Cardiac output not measured: reduced EF not same as reduced CO.  Indeed, low EF may have normal CO at rest and even with mild to moderate exercise.   None of these were tested.  It is also not discussed whether low EF with normal CO changes the impugned pathophysiology that justifies this study.
  3. MoCA not good for finely graded response: gives gross categories.  Coupled with very small cohort and uncontrolled duration and severity of disease, makes it difficult to be confident in the conclusion about maintained cognition or vice versa.
  4. Cerebral hemodynamics:
    1. etCO2 is a highly unreliable surrogate of PaCO2.  NP measures basically provide binary “yes/no” phase of breathing.  The capnograph reading is an unreliable measure of etCO2: the trend may be also unreliable as it has many confounders such as contamination from room air with small changes such as head motion or change over in dominant nostril.  Even if etCO2 were reliably measured one has to account for the low reliability in the relationship between etCO2 and PaCO2 which is the real independent variable. 
    2. VR: the administration of carbogen results in a unknown PaCO2 (Baddeley The British Journal of Radiology, 73 (2000); Peebles J Physiol 1 (2007); hyperventilation in response to carbogen results in unknown PCO2 (see (a)); the slope of Δ MCAV /Δ PCO2 differs above and below baseline.  Basically, without a tightly controlled, repeatable stimulus, precise readings of etCO2, and means to make etCO2 equal to PaCO2, the test-test variability is likely large.  In a small cohort such as this, such method cannot have the sensitivity to detect any but the largest effect size.  So, finding no difference with little difference in cardiac output and grossly imprecise VR test and small cohort is not informative.
  5. I am not sure why baseline etCO2 was measured or its significance. In any event, as mentioned above is poorly representative of PaCO2.  As well the gradient between PaCO2 and etCO2 increases with deadspace which itself increases with worsening HF. 
  6. I am not sure what can reliably be concluded about drug effects when one has such a small study and can’t control for many confounders between those taking and not taking the drugs. 

In summary:  The main conclusion is NVC impaired with low EF may hold.  The other measured parameters, and any relation to drugs and cardiac output, I am not convinced. 

Reviewer 2 Report

This manuscript aims to investigate the association between reduced cardiac output in heart failure and brain hemodynamics. Subjects with HFrEF and HF recovered EF were studied. Control healthy subjects were age and sex matched. Cerebrovascular reactivity was measured using the TCD in response to inhalation of 5% CO2, hyperventilation, and visual stimulation. Authors report that NVC was significantly impaired in HFrEF group. This is a well-written and well-designed study that target an important area of research. There only couple of minor comments that appeared during the revision of this manuscript.
1)The title of the manuscript gives an impression of a methodological paper, which is not the case. I suggest to indicate the result in the title. I also recommend to clarify that patients were with systolic heart failure or heart failure with reduced ejection fraction, in the title.
2) please refer to limitations of the TCD methodology to measure NVC only in the larger brain vessels. There are several methods available that could complement assessment of cerebral hemodynamics and NVC in older adults and in those with CVD including functional near infrared spectroscopy and dynamic retinal vessel analysi (PMID 31209739, PMID 31676966, PMC3363263, etc)

Round 2

Reviewer 1 Report

  1. Introduction: The first paragraph again speaks of cardiac output and then in the last sentence switches to SV.  This is not quite honest as reduced SV is not the same as reduced CO so the mentioned relationships may not apply.  If this is about SV with unknown CO, this should be made clear.  Similarly the second paragraph is not perfectly clear.  Would it not be most honest to say what is said in the last paragraph:  Reduced CO is associated with xxxx, but purpose of the study is to examine the effects of reduced EF alone, with presumed normal CO (good NYHC) on cerebral vascular function as indicated by 3 surrogates parameters….
  2. Methods: etCO2 is mentioned but not what it is used for.  It is also in the tables, but again not why.  It is not in the Discussion.  I would just leave it out.
